# Micromorphological Traits of Balcanic *Micromeria* and Closely Related *Clinopodium* Species (Lamiaceae)

**DOI:** 10.3390/plants10081666

**Published:** 2021-08-13

**Authors:** Dario Kremer, Edith Stabentheiner, Faruk Bogunić, Dalibor Ballian, Eleni Eleftheriadou, Danijela Stešević, Vlado Matevski, Vladimir Ranđelović, Daniella Ivanova, Mirko Ruščić, Valerija Dunkić

**Affiliations:** 1Faculty of Pharmacy and Biochemistry, University of Zagreb, A. Kovačića 1, 10000 Zagreb, Croatia; dkremer@pharma.hr; 2Institute of Biology, Karl-Franzens University, Schubertstrasse 51, 8010 Graz, Austria; edith.stabentheiner@uni-graz.at; 3Faculty of Forestry, University of Sarajevo, Zagrebačka 20, 71000 Sarajevo, Bosnia and Herzegovina; f.bogunic@sfsa.unsa.ba (F.B.); balliandalibor9@gmail.com (D.B.); 4Slovenian Forestry Institute, Večna pot 2, 1000 Ljubljana, Slovenia; 5School of Forestry and Natural Environment, Aristotle University of Thessaloniki, 54124 Thessaloniki, Greece; eelefthe@for.auth.gr; 6Faculty of Natural Sciences and Mathematics, University of Montenegro, Džordža Vašingtona bb, 81 000 Podgorica, Montenegro; denist@t-com.me; 7Faculty of Natural Sciences and Mathematics, Ss. Cyril and Methodius University, Gazi Baba bb, 1000 Skopje, North Macedonia; vladom@pmf.ukim.mk; 8Faculty of Sciences and Mathematics, University of Niš, Višegradska 33, 18000 Niš, Serbia; vladar@pmf.ni.ac.rs; 9Institute of Biodiversity and Ecosystem Research, Bulgarian Academy of Sciences, Acad. Georgi Bonchev Str., bl. 23, 1113 Sofia, Bulgaria; divanova_fern@yahoo.com; 10Faculty of Science, University of Split, Ruđera Boškovića 33, 21000 Split, Croatia; mrus@pmfst.hr

**Keywords:** Balkan Peninsula, capitate trichomes, glandular trichomes, peltate trichomes, pollen, SEM

## Abstract

A study of the trichomes types and distribution and pollen morphology was carried out in nine *Micromeria* taxa (*M. cristata* ssp. *cristata*, *M. cristata* ssp. *kosaninii*, *M. croatica*, *M. graeca* ssp. *graeca*, *M. graeca* ssp. *fruticulosa*, *M. juliana*, *M. kerneri*, *M. longipedunculata* and *M. microphylla*) and five closely related *Clinopodium* species (*C. dalmaticum*, *C. frivaldszkyanum*, *C. pulegium*, *C. serpyllifolium* and *C. thymifolium*) from the Lamiaceae family of the Balkan Peninsula. By scanning electron microscope, non-glandular trichomes, peltate and capitate trichomes were observed on the calyx, leaves and stem of the studied species. Two subtypes of capitate trichomes were observed in *Micromeria* species: subtype 1 (consisting of a basal epidermal cell and an elliptically shaped head cell) and subtype 2 (consisting of a basal epidermal cell, two to three stalk cells and a round head cell). In *Clinopodium* species, three types of capitate trichomes were observed: subtype 1, subtype 3 (consisting of a basal epidermal cell, a short peduncle cell, and a single round head cell), and subtype 4 (consisting of a basal epidermal cell, a stalk cell, and an elongated head cell). These results support the recent transfer of *Micromeria* species from the section *Pseudomelissa* to the genus *Clinopodium.*

## 1. Introduction

The family Lamiaceae is one of the largest families among the *eudicots*, consisting of more than 240 genera and more than 7200 species [1]. Many of species belonging to this family are aromatic and medicinal plants due to the presence of glandular trichomes that produce essential oil. One of the plant groups in this family is the genus *Micromeria* Benth. and closely related species from the genus *Clinopodium* L. The genus *Micromeria* belongs to subtribe *Menthinae* and tribe *Mentheae* (subfam. *Nepetoideae*). According to Briquet [2], Killick [3] and Greuter et al. [4], it is considered part of the indistinctly defined “*Satureja*” complex. However, this opinion is slowly being abandoned today. According to Bentham [5], *Micromeria* species form a separate genus and this opinion prevails today [6,7]. The genus *Micromeria* includes 54 [8] to 70/20 [9] perennial herbs, subshrubs and shrubs, rarely annual herbs. According to Bräuchler et al. [8,10], the range of *Micromeria* species extends from India and China to the Macaronesian Archipelago and from the Mediterranean to South Africa and Madagascar. Bräuchler et al. [10] showed that the genus *Micromeria* is polyphyletic and that a revision of this genus is necessary. In accordance with this opinion, Bräuchler et al. [11] transferred the taxa included to the section *Pseudomelissa* from the genus *Micromeria* to the genus *Clinopodium*.

*Micromeria* species are more or less aromatic plants with opposite leaves and thyrsoid or sometimes racemelike, dense or loose, spikelike inflorescences. The leaf blade is entire or with a few indistinct teeth, often with a thickened and revolute margin. The petiole is distinct, short or minute. The calyx is tubular in the lower part and five-lobed in the upper part, almost actinomorphic to distinctly two-lipped, mostly hairy in the mouth. The upper lip is three-lobed; the lower lip is two-lobed, divided to the base and their lobes are mostly narrower and longer than those of the upper lip. The teeth are ciliate or not. The corolla is strongly two-lipped, with a straight tube. The upper lip is emarginated; the lower lip is three-lobed with middle lobe broader than lateral [8,10,12]. *Clinopodium* species are also aromatic plants with opposite leaves and flowers arranged in opposite, axillary, dense, many-flowered cymes. The calyx is tubular with a curved tube and more or less two-lipped in the upper part, hairy in the mouth. The teeth are almost never ciliate. The corolla is two-lipped, with straight tube. The upper lip is entire or emarginated, while the lower lip is three-lobed with middle lobe wider than lateral [8,10].

The type and distribution of secretory and non-secretory trichomes play an important role in taxonomic study. The study of trichomes in *Micromeria* and *Clinopodium* species has been reviewed by several authors [13,14,15,16,17,18,19]. Palynology also provides valuable taxonomic data and information on the origin and evolution of species [20]. The pollen of *Micromeria* and *Clinopodium* species has also been studied by several authors [15,16,21,22,23,24].

This study represents a continuation of our investigations of Balkan *Micromeria* and *Clinopodium* species. Instead of studying additional species individually, we included all available *Micromeria* and *Clinopodium* species from Balkan Peninsula in this paper. Of the genus *Clinopodium*, only species transferred from *Micromeria* sect. *Pseudomelissa* were included in this study. So, the aim of this study is to gain insight into the micromorphological traits of *Micromeria* species as well as the closely related *Clinopodium* species recently transferred from the *Micromeria* sect. *Pseudomelissa*. The main objective is to determine whether micromorphological traits support the transfer of species from section *Pseudomelissa* to the genus *Clinopodium*.

## 2. Results and Discussion

### 2.1. Trichomes

The non-glandular and glandular trichomes were observed in the studied *Micromeria* and *Clinopodium* species. Non-glandular (NG) trichomes have been observed on the leaves, calyces, and stem of both genera. These trichomes are unbranched, uniseriate, bi-cellular to multicellular and most often bent on different sides. They could be called attenuated trichomes because they gradually taper from base to tip [25]. The length of the NG trichomes is highly variable and ranges from very short trichomes, especially on the adaxial leaf surface, to very long hairs on the calyx and stem (Figure 1 and Figure 2). The surface of these trichomes is warty (Figure 1). The main function of NG trichomes is to protect the plant from water loss. The same type of NG trichomes is present in both *Micromeria* and *Clinopodium* species.

The occurrence and frequency of NG trichomes on the adaxial and abaxial sides of the leaf, the outer side of the calyx and the stem are shown in Table 1 and Table 2. All *Micromeria* species studied have a more or less abundant coverage of NG trichomes on the adaxial and abaxial leaf surface (Figure 1). A certain exception is *M. longipedunculata*, in which the upper and lower leaf surfaces are less hairy. The adaxial and abaxial leaf surfaces and stem of *M. graeca* ssp. *fruticulosa* and the stem of *M. cristata* have the highest number of long NG trichomes forming an abundant covering.

In general, *Micromeria* species have a denser indumentum of attenuated trichomes on leaves than *Clinopodium* species. *Clinopodium serpyllifolium* is an exception as it has the densest NG trichomes cover of all *Micromeria* and *Clinopodium* species examined (Figure 2). Studies by Firat et al. [22] also showed very dense indumentum on the leaves and stem of *Clinopodium serpyllifolium* ssp. *sirnakense*. The frequency of NG trichomes on the calyx of *Clinopodium* species is similar to this coverage of NG trichomes in *Micromeria* species. A slightly lower frequency of NG trichomes was observed on the calyx of *C. frivaldszkyanum* and *C. pulegium*, and on the stem of *C. frivaldszkyanum*. The presence of NG trichomes has been previously well documented in *Micromeria* [13,15,16] and *Clinopodium* [13,14,17,18] species, as well as in other Lamiaceae [13,14,26].

Glandular trichomes were observed on all plant parts of the *Micromeria* and *Clinopodium* species examined. These trichomes can be further divided into two main subtypes, namely, peltate and capitate trichomes. Peltate trichomes consist of a basal cell, a very short unicellular stalk and a multicellular head with a large subcuticular space (Figure 1). These trichomes are also known from some other Lamiaceae species [13,26,27,28,29,30,31,32]. The distribution and frequency of peltate trichomes on the studied plant parts are shown in Table 1 and Table 2. Although peltate trichomes are present on all plant parts studied, they are more abundant on the abaxial leaf surface and on the calyx. They are completely absent on the adaxial side of the leaf in *M. croatica* and *M. longipedunculata* (Table 1). Peltate trichomes were previously reported in *Micromeria fruticosa* [13], *Micromeria myrtifolia* Boiss. et Hohen. [33], *M. croatica* [15], *M. kerneri*, *M. juliana* [16] and *M. longipedunculata* [34]. Husain et al. [14] showed the presence of peltate trichomes on the leaves of *Clinopodium serpyllifolium*, *C. dalmaticum* and *C. thymifolium*. Peltate trichomes were also detected in micrographs of *C. thymifolium* leaves presented by Marin et al. [35]. Mladenova et al. [19] observed these trichomes in the leaves of *C. frivaldszkyanum*, although they did not describe them as peltate.

The capitate trichomes can be further divided into several subtypes. Subtype 1 (C1) capitate trichomes consist of a basal epidermal cell and an elliptically shaped head cell with a larger subcuticular space. C1 trichomes are not erect but can be described as adherent to the surface (Figure 1). This type was observed on both the abaxial and adaxial sides of the leaves, on the stem and on the outer side of the calyx. They are present in all *Micromeria* and *Clinopodium* species studied (Table 1 and Table 2). These trichomes were described in detail in *Micromeria croatica* by Kremer et al. [15].

C1 trichomes have also been observed in *M. kerneri* and *M. juliana* [16]. Although Husain et al. [14] did not mention C1 trichomes in *M. longipedunculata*, unclear SEM micrographs probably show just the C1 type of trichomes. On the other hand, SEM micrographs of *M. fruticosa* presented by Werker et al. [13] did not show C1 trichomes. According to drawings made by Koca [33], it seems that C1 trichomes are also present in *Micromeria myrtifolia.* C1 trichomes were also observed in *Clinopodium dalmaticum*, *C*. *pulegium*, *C*. *serpyllifolium* and *C*. *thymifolium* [17].

Hanlidou et al. [36] described trichomes comparable to C1 trichomes in *Calamintha menthifolia* Host as short and usually curved trichomes. These trichomes are regularly present in all studied *Micromeria* and *Clinopodium* species, but they cannot be considered specific only to these two genera. In addition, the SEM micrographs presented by Werker et al. [13] show C1 trichomes in *Majorana syriaca* (L.) Rafin., *Satureja thymbra* L., and *Thymus capitatus* (L.) Hoffmanns. et Link. However, it is evident that C1 trichomes are not as common as peltate trichomes in Lamiaceae species.

Subtype 2 capitate trichomes (C2) are composed of a basal epidermal cell, two to three stalk cells and a rounded head cell with subcuticular space (Figure 1). C2 trichomes are upright while the height of these trichomes varies from short trichomes, mainly on the abaxial side of the leaf, to quite long trichomes on the calyx and stem. However, they are often absent on the adaxial leaf side. C2 trichomes are observed only in *Micromeria* species studied.

So far, these trichomes have been observed in *Micromeria fruticosa* [13], *M. myrtifolia* [34], *M. croatica* [15], *M. kerneri*, *M. juliana* [16], and *M. longipedunculata* [34]. These trichomes are more common than C1 trichomes in Lamiaceae species and trichomes comparable to C2 trichomes have also been described in *Salvia* L. [30] and *Satureja* L. [37,38,39] species.

Subtype 3 capitate trichomes (C3) are also erect and consist of a basal epidermal cell, a relatively short stalk cell, and a single roundish cell head with subcuticular space. Although C3 trichomes were observed in all investigated *Clinopodium* species, they were almost absent in *C. frivaldszkyanum*, *C. pulegium* and *C. thymifolium* (Table 2). Marin et al. [35] did not mention C3 trichomes, but they are visible in the presented SEM micrographs of *C. thymifolium* leaves. They are also present in the micrographs of *Satureja montana* L., *S*. *subspicata* Vis. and *S*. *kitaibelii* Wierzb. ex Heuff. presented by Dunkić et al. [38] and Dodoš et al. [39].

The capitate trichomes of subtype 4 (C4) are upright and consist of a basal epidermal cell, a stalk cell and an elongated head cell. In these trichomes, the head cell is as narrow as the stalk cells, and only slightly enlarged above, with a subcuticular space. They resemble a finger (Figure 2). Although C4 trichomes are present in all *Clinopodium* species studied, they are relatively rare trichomes (Table 2). The presence of C4 trichomes was previously observed in *Clinopodium dalmaticum*, *C. pulegium*, *C*. *serpyllifolium* and *C*. *thymifolium* by Dunkić et al. [17]. These trichomes are also visible in SEM micrographs of leaves of *Micromeria fruticosa* and *Clinopodium thymifolium* presented by Werker et al. [13] and Marin et al. [35], respectively. C4 trichomes are visible in SEM micrographs of *Calamintha menthifolia*, and in micrographs of *Satureja montana* and *S*. *subspicata* presented by Hanlidou et al. [36] and Dunkić et al. [38], respectively. Unclear micrographs presented by Al-Zubaidy et al. [40] probably show just C4 trichomes in *Clinopodium vulgare* L. ssp. *vulgare* and *C*. *vulgare* ssp. *arundanum* Boiss. In addition, they can be observed in SEM micrographs of *Majorana syriaca*, *Salvia fruticosa* Mill., *S*. *officinalis* L. [13], and *S*. *divinorum* Epling et Játiva [30]. Although data from the literature [13] suggest that C4 trichomes may sometimes be present in *Micromeria* species, our results show that they are characteristic for *Clinopodium* species.

The results presented here show that NG trichomes, peltate trichomes and capitate trichomes of subtype 1 are present in both *Micromeria* and *Clinopodium* taxa. On the other hand, the capitate trichomes of subtype 2 are present only in studied *Micromeria* taxa, while subtypes 3 and 4 are present only in *Clinopodium* taxa. Since the investigated taxa of the genus *Clinopodium* belong to the former *Micromeria* sect. *Pseudomelissa*, it can be concluded that micromorphological traits also support the recent transfer of *Micromeria* sect. *Pseudomelissa* to the genus *Clinopodium*.

### 2.2. Pollen

The pollen of all *Micromeria* and *Clinopodium* species examined is single (monad pollen) and isopolar with an elliptical equatorial outline (Figure 3 and Figure 4).

The polar view shows a circular shape with visible ends of apertures. The pollen has six apertures (hexacolpate pollen) located in the equatorial pollen belt (zonocolpate pollen). The apertures are long and rather narrow, widest in the middle and gradually narrowing towards the poles. The margins of the apertures are clear and sharp, while the ends are narrow and pointed. The membranes are ornamented. The apocolpium is relatively small, while the mesocolpium is quite large. The pollen exine is semitectate with medium reticulate ornamentation. The reticulum meshes are unequal (heterobrochate reticulum type) with more or less smooth surfaces of muri. Only the pollen exine of *M. cristata* ssp. *cristata* and *M. cristata* ssp. *kosaninii* show uneven surfaces of the muri (Figure 3). The lumina vary in size and are narrower or about the same width as the muri, rarely wider than muri as in *M. croatica* (Figure 3). The shape of the lumina is irregular with obtuse angles. So far, the pollen of *Micromeria marginata*, *M. croatica*, *M. longipedunculata*, and *M. imbricata* (Forssk.) C. Chr. have been described in detail [15,21,23]. In general, there is no visible difference in pollen shape between the *Micromeria* and *Clinopodium* species studied.

The size of pollen grains according to polar and equatorial axis is shown in Table 3. According to Erdtman [20], the pollen of *M. cristata* ssp. *kosaninii*, *M.*
*longipedunculata*, *M. microphylla*, and *C. thymifolium* belong to the small pollen, while the pollen of the other investigated species belong to the medium pollen. *Micromeria longipedunculata* and *M. microphylla* have the smallest pollen (according to the length of the longer axis, 21.10 and 21.80 µm, respectively). On the other hand, *M. graeca* ssp. *graeca* and *M. graeca* ssp. *fruticulosa* have the largest pollen (by the longer axis, 34.65 and 33.86 µm, respectively). The size of pollen grains of *M. croatica* in this study was similar to the results of Kremer et al. [15], who studied the pollen of this species from another locality. Comparable results were also obtained for the pollen of *M. juliana*, *M. kerneri*, and *M. longipedunculata* [16,34]. *Micromeria marginata* has middle-large pollen with a polar axis of 26.6 μm and an equatorial diameter of 35.2 μm [21]. According to Doaigey et al. [23], the pollen of *Micromeria imbricata* has a length of 34.57 and 31.33 µm (polar and equatorial axis, respectively). The smallest pollen among the investigated *Clinopodium* species was in *C. thymifolium* (23.52 µm), while the largest pollen (according to longer axis 32.47 µm) was in *C. dalmaticum*. The measure of polar length of *Clinopodium foliosum* pollen ranges from 20.45–28.6 μm and for *C. menthifolium* from 25.7–31.4 μm [24]. On the other hand, the equatorial width is 20.46–28.57 μm for *C. foliosum* and 20.9–31.43 μm for *C. menthifolium* [24].

The results of the ANOVA show the significant difference between most of the studied taxa in terms of pollen size (Table 4 and Table 5). Although there is a significant difference between most of *Micromeria* and *Clinopodium* taxa, this difference is not clear. There is a significant difference between the closely related taxa *M. cristata* ssp. *cristata* and M. *cristata* ssp. *kosaninii* in terms of equatorial diameter. However, there is a significant difference between the closely related taxa *M. graeca* ssp. *graeca* and *M. graeca* ssp. *fruticulosa* according to the polar axis. There is no significant difference between *M. croatica* and *M*. *pseudocroatica*, which are one species according to Bräuchler et al. [8]. On the other hand, there is a significant difference between *C. dalmaticum* from Orjen Mt (Montenegro) and *C. dalmaticum* from Mesta River Valley (Bulgaria) for both polar and equatorial axes.

## 3. Materials and Methods

### 3.1. Plant Material

The survey included, depending on the point of view [8,9,12,41,42,43,44,45], nine to ten *Micromeria* taxa from the Balkan Peninsula (Table 6, Figure 5, Figure 6 and Figure 7): *M. cristata* (Hampe) Griseb. ssp. *cristata*, *M. cristata* ssp. *kosaninii* (Šilić) Bräuchler et Govaerts (syn. *M. kosaninii* Šilić), *M. croatica* (Pers.) Schott (including *M*. *pseudocroatica* Šilić)*, M. graeca* (L.) Benth. ex Rchb. ssp. *graeca*, *M. graeca* ssp. *fruticulosa* (Bertol.) Guinea (syn. *M*. *fruticulosa* (Bertol.) Šilić), *M. juliana* (L.) Benth. ex Rchb., *M. kerneri* Murb., *M. longipedunculata* Bräuchler, (syn. *M. parviflora* Rchb.), and *M. microphylla* (d’Urv.) Benth. In addition, five to six, depending on point of view [8,12,19,41,42,43,44,45,46], taxa recently transferred from the genus *Micromeria* to the genus *Clinopodium* were studied, namely *Clinopodium dalmaticum* (Benth.) Bräuchler et Heubl (syn. *Micromeria dalmatica* Benth.) including *M. bulgarica* (Velen.) Vandas, *C. frivaldszkyanum* (Degen) Bräuchler et Heubl (syn. *M. frivaldszkyana* (Degen) Velen.), *C*. *pulegium* (Rochel) Bräuchler (syn. *M*. *pulegium* (Rochel) Benth.), *C*. *serpyllifolium* (M. Bieb.) Kuntze (syn. *M. albanica* (Griseb. ex K. Malý) Šilić), and *C*. *thymifolium* (Scop.) Kuntze (syn. *M*. *thymifolia* (Scop.) Fritch.). Each locality was described with GPS coordinates and elevation. Voucher specimens of the plant material were deposited in the Herbarium “Fran Kušan” (HFK-HR), Faculty of Pharmacy and Biochemistry, University of Zagreb, Croatia (Table 6 and Table 7).

Samples of the stems, leaves, and flowers of ten plants per population were fixed in FAA (formalin/96% ethanol/acetic acid/water: 5/70/5/20). After three days the samples were transferred from the fixation medium to 70% (*v*/*v*) ethanol and stored in the refrigerator until analysis.

The samples for scanning electron microscopic (SEM) investigation were transferred from 70% (*v*/*v*) ethanol to 70% (*v*/*v*) acetone. Then, the samples were dehydrated from 70% (*v*/*v*) to 90% (*v*/*v*), and 100% (*v*/*v*) acetone. The dehydrated samples were subjected to critical point drying using CO_2_ as the drying medium (CPD030; Bal-tec, Balzers, Liechtenstein). The samples were then sputter coated with gold (Sputter Coater, Agar Scientific Ltd., Essex, UK) and examined under the scanning electron microscope XL30 ESEM (FEI) with an acceleration voltages of 20 kV in high vacuum mode. The presence and abundance of the different trichome types was qualitatively assessed (− missing, ± rare, + present, ++ abundant, +++ extremely abundant). Pollen was collected from the anthers of several flowers per plant after critical point drying. A total of pollen from ten plants per population studied was mixed and examined under a scanning electron microscope. The length of 30 pollen grains per population was measured. Pollen size is expressed as the length of the longest axis according to Erdman [20]. Common terminology was used in describing trichomes and pollen [25,47].

### 3.2. Statistical Analysis

Descriptive statistics: minimum (Min), maximum (Max), mean, standard deviation (Stdev) and coefficient of variation (CV) for length and diameter of pollen grains were calculated. Polar end equatorial axis lengths of pollen grains were subjected to One-way Analysis of Variance (ANOVA). Differences between taxa were tested with Tukey’s HSD post hoc tests [48].

## 4. Conclusions

Non-glandular trichomes, peltate trichomes, and four subtypes of capitate trichomes were observed on the aerial parts of the *Micromeria* and *Clinopodium* taxa examined. The NG trichomes, peltate trichomes and capitate trichomes of subtype 1 (bent trichomes consisting of a basal epidermal cell and an elliptically shaped head cell) were observed in both *Micromeria* and *Clinopodium* taxa. Capitate trichomes of subtype 2 (erect trichomes consisting of a basal epidermal cell, two to three stalk cells and a rounded head cell) were observed only in *Micromeria* taxa, while capitate trichomes of subtype 3 (upright trichomes consisting of a basal epidermal cell, a relatively short stalk cell and a rounded celled head) and subtype 4 (upright trichomes consisting of a basal epidermal cell, a stalk cell, and a narrow and elongated head cell) were observed only in *Clinopodium* taxa. Such a distribution of capitate trichomes subtypes in *Micromeria* and *Clinopodium* taxa shows that, on the basis of micromorphological traits, it is possible to distinguish taxa from the former *Pseudomelissa* section from other *Micromeria* taxa. In this way, micromorphological traits support the recent transfer of *Micromeria* species from the section *Pseudomelissa* to the genus *Clinopodium*.

On the other hand, the pollen studies have shown that there are no significant differences between pollen of the *Micromeria* and *Clinopodium* taxa.

## Figures and Tables

**Figure 1 plants-10-01666-f001:**
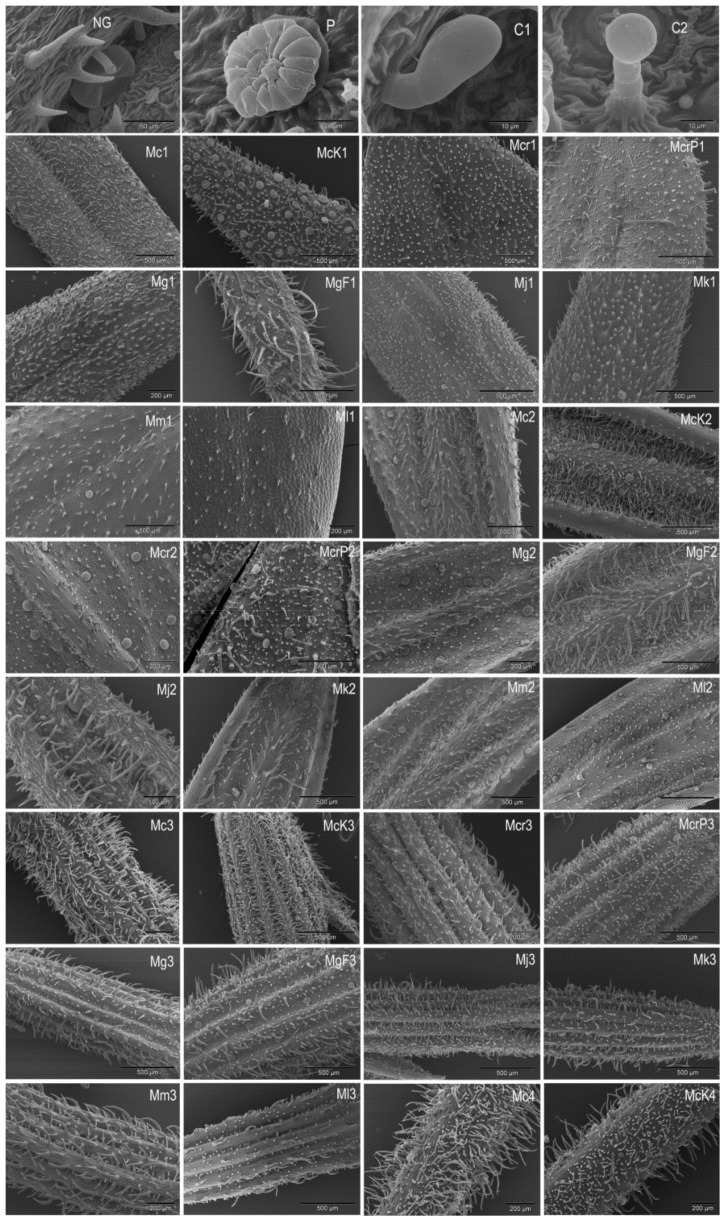
SEM micrographs with different types and distribution of trichomes in *Micromeria cristata* ssp. *cristata* (Mc), *M*. *cristata* ssp. *kosaninii* (McK), *M*. *croatica* (Mcr) including *M*. *pseudocroatica* (McrP), *M*. *graeca* ssp. *graeca* (Mg), *M*. *graeca* ssp. *fruticulosa* (MgF), *M. juliana* (Mj), *M. kerneri* (Mk), *M. microphylla* (Mm), and *M. longipedunculata* (Ml). Non-glandular trichomes (NG), peltate trichome (P), Subtype 1 capitate trichomes (C1) and Subtype 2 capitate trichomes (C2) on the adaxial (1) and abaxial (2) leaf surface, on the calyx (3) and stem (4).

**Figure 2 plants-10-01666-f002:**
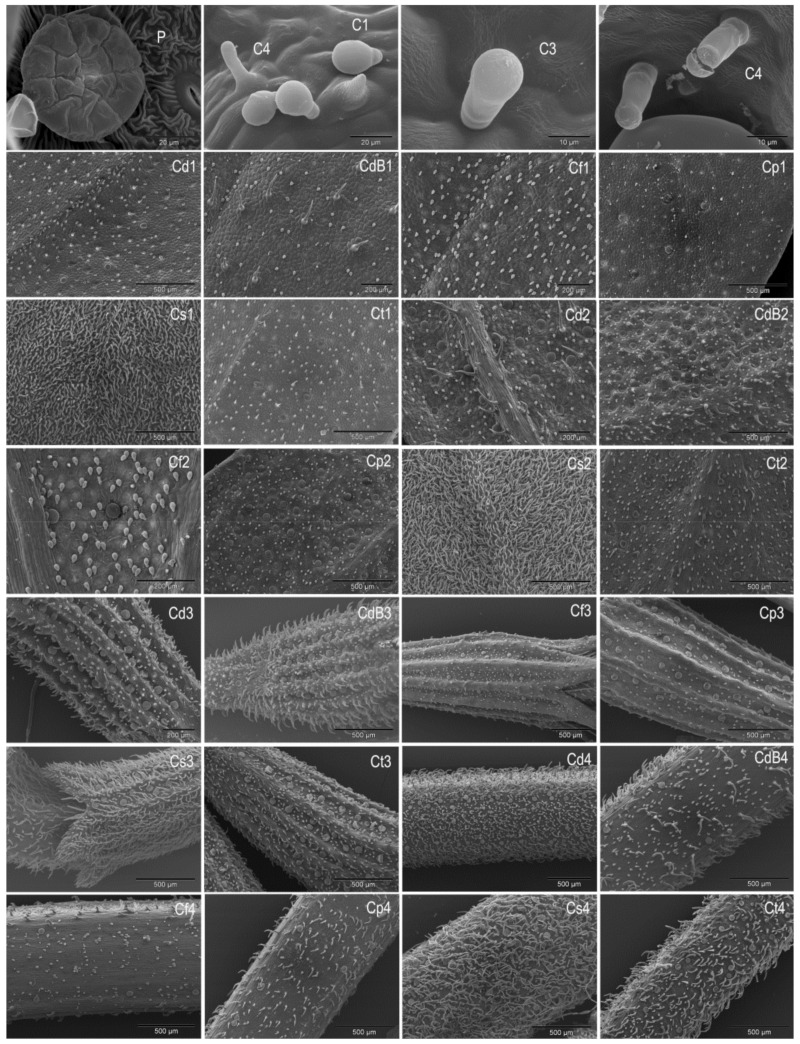
SEM micrographs with different types and distribution of trichomes in *Clinopodium dalmaticum* (Cd) including *Micromeria bulgarica* (CdB), *C*. *frivaldszkyanum* (Cf), *C*. *pulegium* (Cp), *C*. *serpyllifolium* (Cs), and C. *thymifolium* (Ct). Peltate trichome (P), Subtype 1 capitate trichomes (C1), Subtype 3 capitate trichomes (C3) and Subtype 4 capitate trichomes (C4) on the adaxial (1) and abaxial (2) leaf surface, calyx (3), and stem (4).

**Figure 3 plants-10-01666-f003:**
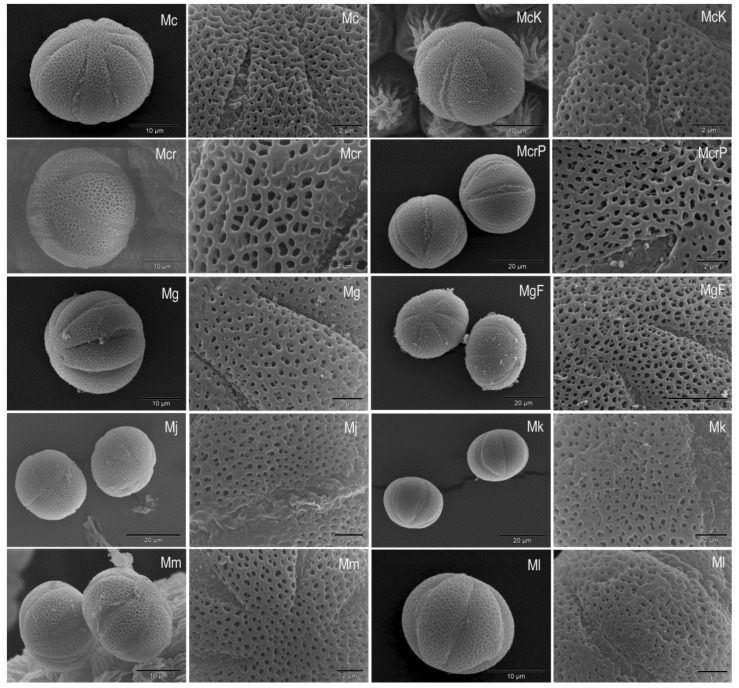
SEM micrographs of pollen grains and exine surface after critical point drying in *Micromeria cristata* ssp. *cristata* (Mc), *M. cristata* ssp. *kosaninii* (McK), *M*. *croatica* (Mcr) including *M*. *pseudocroatica* (McrP), *M. graeca* ssp. *graeca* (Mg), *M. graeca* ssp. *fruticulosa* (MgF), *M. juliana* (Mj), *M. kerneri* (Mk), *M. microphylla* (Mm), and *M. longipedunculata* (Ml).

**Figure 4 plants-10-01666-f004:**
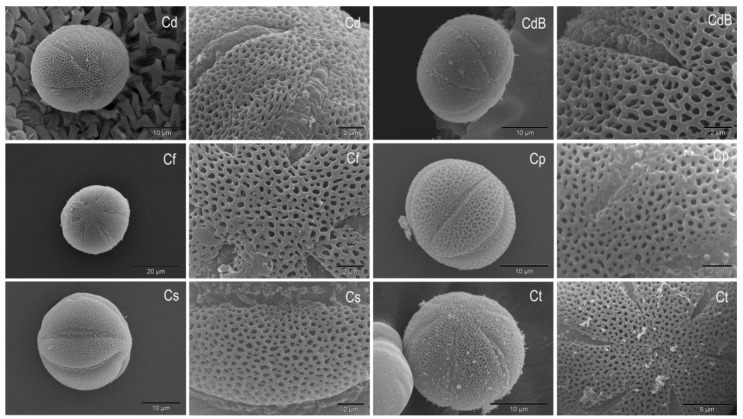
SEM micrographs of pollen grains and exine surface after critical point drying in *Clinopodium dalmaticum* (Cd) including *Micromeria bulgarica* (CdB), *C. frivaldszkyanum* (Cf), *C. pulegium* (Cp), *C. serpyllifolium* (Cs), and *C. thymifolium* (Ct).

**Figure 5 plants-10-01666-f005:**
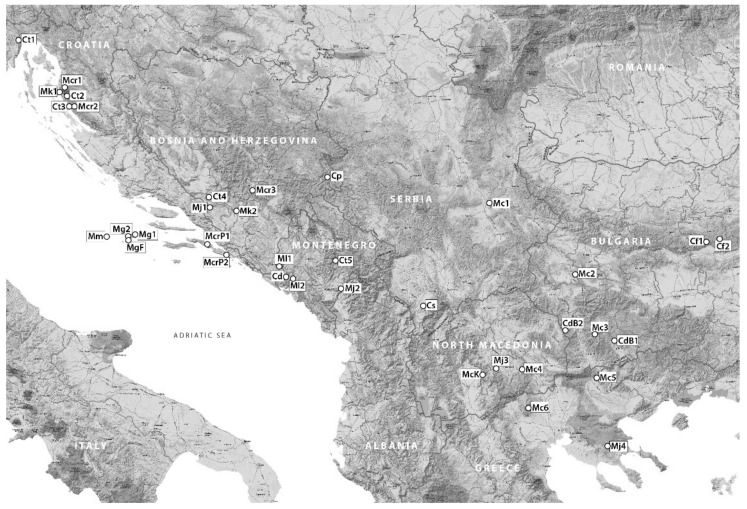
Collection sites of studied *Micromeria* and *Clinopodium* taxa: *M. cristata* ssp. *cristata* (Mc), *M. cristata* ssp. *kosaninii* (McK), *M*. *croatica* (Mcr) including *M*. *pseudocroatica* (McrP), *M. graeca* ssp. *graeca* (Mg), *M. graeca* ssp. *fruticulosa* (MgF), *M. juliana* (Mj), *M. kerneri* (Mk), *M. microphylla* (Mm), *M. longipedunculata* (Ml), *C. dalmaticum* (Cd) including *M. bulgarica* (CdB), *C. frivaldszkyanum* (Cf), *C. pulegium* (Cp), *C. serpyllifolium* (Cs), and *C. thymifolium* (Ct).

**Figure 6 plants-10-01666-f006:**
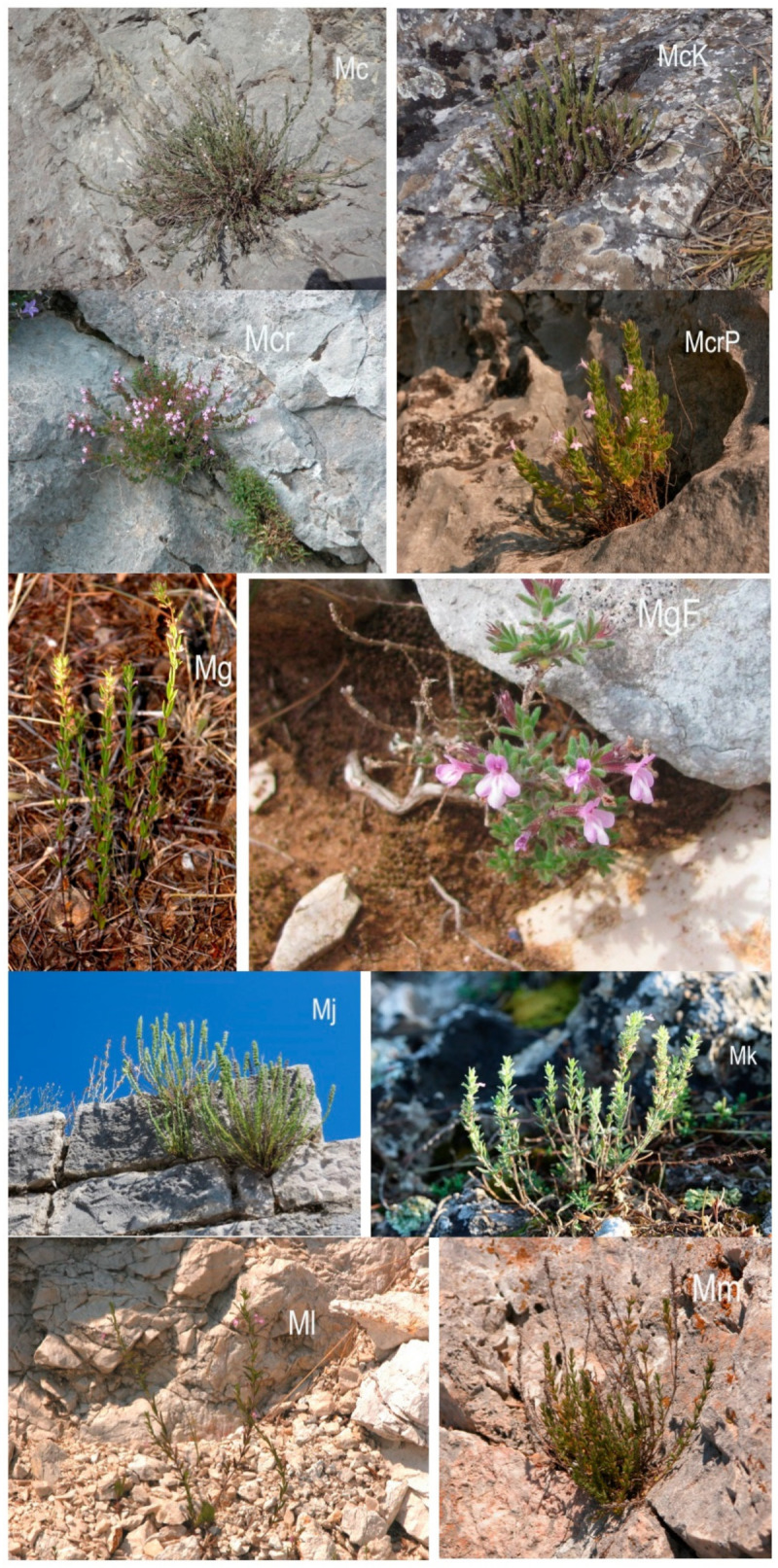
Photographs of *Micromeria cristata* ssp. *cristata* (Mc), *M*. *cristata* ssp. *kosaninii* (McK), *M*. *croatica* (Mcr) including *M*. *pseudocroatica* (McrP), *M*. *graeca* ssp. *graeca* (Mg), *M*. *graeca* ssp. *fruticulosa* (MgF), *M. juliana* (Mj), *M. kerneri* (Mk), *M. longipedunculata* (Ml) and *M. microphylla* (Mm).

**Figure 7 plants-10-01666-f007:**
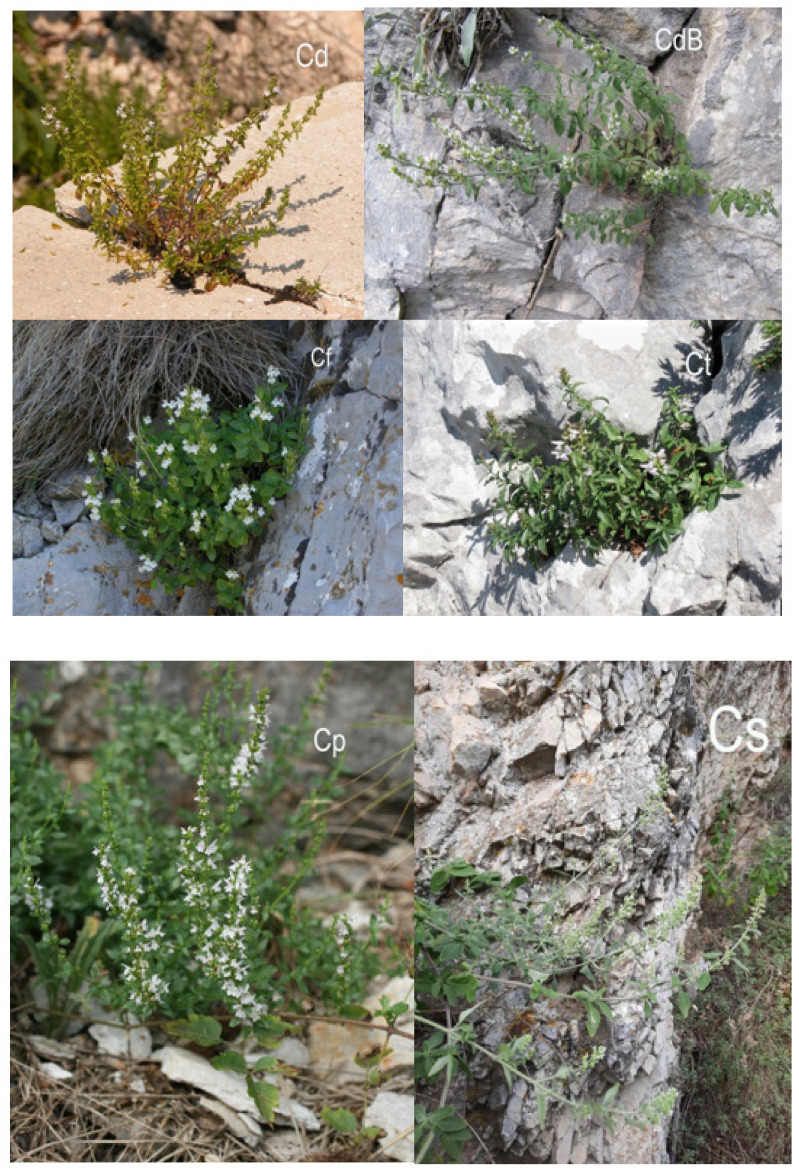
Photographs of *Clinopodium dalmaticum* (Cd) including *Micromeria bulgarica* (CdB), *C. frivaldszkyanum* (Cf), *C. thymifolium* (Ct), *C. pulegium* (Cp) and *C. serpyllifolium* (Cs).

**Table 1 plants-10-01666-t001:** Occurrence and frequency of trichomes on aerial parts of *Micromeria* taxa.

Taxon/	Trichomes	Leaf		Calyx	Stem
Locality		Adaxial	Abaxial		
*M. cristata*	NG	+	+/++	++	+
ssp. *cristata*	peltate	+	+	++	+
Humsko	capitate C1	+/++	+	+	+
Brdo	capitate C2	−	±/+	+	+
*M. cristata*	NG	+	+/++	++	++
ssp. *cristata*	peltate	+	+	+	+
Vitosha Mt	capitate C1	+	+	+	+
	capitate C2	−	±/+	+	±
*M. cristata*	NG	+	+/++	++	+/++
ssp. *cristata*	peltate	+/++	+	+/++	+
Pirin Mt	capitate C1	+	+	+	+
	capitate C2	−	+	±	±
*M. cristata*	NG	+/++	+/++	+/++	+
ssp. *cristata*	peltate	+	+	+	+
Demir	capitate C1	±/+	+	−/±	+
Kapija	capitate C2	−/±	+/++	+/++	±/+
*M. cristata*	NG	+/++	+	++	+
ssp. *cristata*	peltate	+	+	+	+
Nomos	capitate C1	±/+	+	+	+
Serron	capitate C2	−	+	+	+/++
*M. cristata*	NG	+	+/++	++	+/++
ssp. *cristata*	peltate	+	+	+	±/+
Paikon Mt	capitate C1	+	+/++	+	+
	capitate C2	−	+/++	+	+/++
*M. cristata*	NG	++	++	++	+/++
ssp. *kosaninii*	peltate	+/++	+	+	+
Pletvar	capitate C1	±/+	±/+	+	+
	capitate C2	±/+	+/++	±/+/++	+
*M. croatica*	NG	+	+	+	+
Rossijev	peltate	−	+/++	+	+
kuk	capitate C1	−	±/+	±	−
	capitate C2	±/++	++	++	−
*M. croatica*	NG	+	+	++	+
Stupačinovo	peltate	−	+	−/±	+
	capitate C1	−	+/++	+	±
	capitate C2	−	±/++	±	+/++
*M. croatica*	NG	+	+	++	+
Dubočani	peltate	−	++	+/++	+
	capitate C1	−	±/++	+/++	±
	capitate C2	−/++	±/++	±/+	+/++
*M. croatica*	NG	+	+	+	+
(syn. *M*.	peltate	−	++	+	+
*pseudocroatica*)	capitate C1	±	+/++	+	+
Pijavičino	capitate C2	±	++	++	+/++
*M. croatica*	NG	+	+	+	+
(syn. *M*.	peltate	−	++	+	+
*pseudocroatica*)	capitate C1	±	+/++	+	+
Prapratno	capitate C2	±	++	++	+/++
*M. graeca*	NG	++	++	+/++	+/++
ssp. *graeca*	peltate	+	+	+	+
Malo	capitate C1	±	++	++	+
zlo polje	capitate C2	±	+	++	+
*M. graeca*	NG	++	++	+/++	+/++
ssp. *graeca*	peltate	+	++	+	+
Komiža	capitate C1	±	++	+	+
	capitate C2	±	+	++	±
*M. graeca*	NG	++	+	+/++	++
ssp. *fruticulosa*	peltate	±/++	+	+	+
Sušac	capitate C1	±	+	+	+
	capitate C2	±	+	+	+
*M. juliana*	NG	+	+	++	+/++
Grude	peltate	+	+	+	+
	capitate C1	±	+	+	±/+
	capitate C2	−	+	+	+
*M. juliana*	NG	+	+	++	+
Cijevna	peltate	+	+	+	+
	capitate C1	+	±	++	+
	capitate C2	−	++	++	+
*M. juliana*	NG	++	++	++	++
Rajec Reka	peltate	+	+	+	+
	capitate C1	±	+	+	+
	capitate C2	−	+	+	+
*M. juliana*	NG	+	+/++	+/++	+/++
Cholomon Mt	peltate	+	+	±/+	+
	capitate C1	+	+	+	+
	capitate C2	−	±/+	±	+
*M. kerneri*	NG	+/++	+	++	+/++
Zavratnica	peltate	+	+	+/++	+
bay	capitate C1	±	+	±	+
	capitate C2	−	+	++	+
*M. kerneri*	NG	+/++	+	++	+
Mostar	peltate	±	+	+	+
	capitate C1	±	+	±/+	±
	capitate C2	−	±	++	−
*M.*	NG	+	+	+	+/++
*longipedunculata*	peltate	−	+	+	±
Jazina	capitate C1	±	+/++	+	+
	capitate C2	−	+	+	±
*M.*	NG	+	+	+	+
*longipedunculata*	peltate	−	+	+	±
Krivošije Mt	capitate C1	±	+/++	+	+
	capitate C2	−	+	+	±/+
*M*. *microphylla*	NG	+	+	++	+
Svetac Island	peltate	+	+	+	+
	capitate C1	−/±/+	±/+	±	±/+
	capitate C2	−	+	+/++	+

Note: NG = non-glandular trichomes; trichomes frequency: −, trichomes completely missing; ±, trichomes are present in small numbers; +, trichomes are moderately present; ++, trichomes are present in large numbers.

**Table 2 plants-10-01666-t002:** Occurrence and frequency of trichomes on aerial parts of *Clinopodium* taxa.

Taxon/	Trichomes	Leaf		Calyx	Stem
Locality		Adaxial	Abaxial		
*C. dalmaticum*	NG	+	+/++	+	+/++
Orjen	peltate	+	+/+	+	+
	capitate C1	+	±/+	±/+	+/++
	capitate C3	±	++	+/++	−
	capitate C4	−	±	±	±
*C. dalmaticum*	NG	+	+/++	+/++	+
(syn. *M. bulgarica*)	peltate	±	+	+/++	+
Mesta	capitate C1	+	+/++	±/+/++	++
River	capitate C3	−	−/±	±/+	±
Valley	capitate C4	−	−	±/+	−/±
*C. dalmaticum*	NG	+	+	+/++	+
(syn. *M. bulgarica*)	peltate	±	++	+/++	+
Vlahina Mt	capitate C1	+	+/++	+	++
	capitate C3	−	−	−	−
	capitate C4	−	−	−/±	−/±
*C. frivaldszkyanum*	NG	±	±	+	−/+
Malusha peak	peltate	±/+	+	+/++	−/+
	capitate C1	+	++	+	+
	capitate C3	−	−	−	−
	capitate C4	−	−/±	+	−
*C. frivaldszkyanum*	NG	±	±/+	+	+
Vikanata Skala	peltate	±	+	+	+
Nature	capitate C1	+	+	+	+
Monument	capitate C3	−	−	−/±	−
	capitate C4	−	−	+	−
*C* *. pulegium*	NG	±	+	+	+
Međeđa	peltate	+	++	++	+
	capitate C1	±	+/++	+/++	+
	capitate C3	−	−	−	−
	capitate C4	−	−	±	−
*C*. *serpyllifolium*	NG	+++	+++	++	++
Prizren	peltate	+	+	+	+
	capitate C1	+	++	++	+/++
	capitate C3	−	±	±	±
	capitate C4	−	±	±	−
*C* *. thymifolium*	NG	±	+	+	+
Učka Mt	peltate	+	+/++	+/++	+
	capitate C1	+/++	+/++	+/++	+/++
	capitate C3	−	−	±	−
	capitate C4	−	−	±	−
*C* *. thymifolium*	NG	+	+	+	+/++
Dokozina	peltate	+	+/++	+/++	+
plan	capitate C1	+	++	++	+
	capitate C3	−	−	−	−
	capitate C4	−	−	−	−
*C* *. thymifolium*	NG	+	+	+	+
Šušanj	peltate	+	++	+/++	+
	capitate C1	+	++	+/++	+
	capitate C3	−	−	−	−
	capitate C4	−	−	±	−
*C* *. thymifolium*	NG	+	+	+	+
Blidinje	peltate	+	+/++	+/++	+
	capitate C1	+	++	+/++	+
	capitate C3	±	−	−	−
	capitate C4	−	−	−	−
*C* *. thymifolium*	NG	±	+	+	±/+
Manastir	peltate	+	+/++	+/++	+
Morače	capitate C1	+/++	++	++	+/++
	capitate C3	−	−	−	−
	capitate C4	−	−	±	−

Note: NG = non-glandular trichomes; trichomes frequency: −, trichomes completely missing; ±, trichomes are present in small numbers; +, trichomes are moderately present; ++, trichomes are present in large numbers; +++, trichomes completely cover the surface.

**Table 3 plants-10-01666-t003:** Descriptive statistics for length of *Micromeria* and *Clinopodium* pollen grain according to polar (P) and equatorial (E) axis. Mean, Stdev, Min, and Max are in µm; CV is in %.

Taxon		Polar Axis (P)	Equatorial Axis (E)	Pollen Size	P/E Ratio
*M. cristata* ssp. *cristata*	Mean	21.65	25.90	Medium	0.84
	Stdev	1.10	2.28		
	Min	19.40	21.30		
	Max	23.60	28.60		
	CV	5.08	8.80		
*M. cristata* ssp. *kosaninii*	Mean	21.00	22.94	Small	0.92
	Stdev	0.91	1.10		
	Min	19.00	21.30		
	Max	22.70	25.00		
	CV	4.33	4.80		
*M*. *croatica*	Mean	25.97	28.22	Medium	0.92
	Stdev	1.86	1.93		
	Min	23.90	25.70		
	Max	30.50	32.40		
	CV	7.16	6.84		
*M*. *croatica* (syn. *M*. *pseudocroatica*)	Mean	26.35	29.65	Medium	0.89
	Stdev	2.38	2.80		
	Min	22.10	25.40		
	Max	30.00	23.90		
	CV	9.03	9.44		
*M. graeca* ssp. *graeca*	Mean	28.65	34.65	Medium	0.83
	Stdev	2.42	2.69		
	Min	24.60	31.20		
	Max	33.00	44.00		
	CV	8.45	7.76		
*M*. *graeca* ssp. *fruticulosa*	Mean	26.73	33.86	Medium	0.79
	Stdev	3.16	2.74		
	Min	22.40	28.10		
	Max	35.20	38.00		
	CV	11.82	8.09		
*M. juliana*	Mean	23.37	27.10	Medium	0.86
	Stdev	1.21	1.29		
	Min	21.30	25.10		
	Max	26.30	29.40		
	CV	5.18	4.76		
*M. kerneri*	Mean	23.92	27.92	Medium	0.86
	Stdev	1.40	1.02		
	Min	21.90	26.50		
	Max	26.20	29.90		
	CV	5.85	3.65		
*M*. *longipedunculata*	Mean	18.18	21.10	Small	0.86
	Stdev	0.74	1.67		
	Min	17.10	18.30		
	Max	19.40	25.50		
	CV	4.07	7.91		
*M. microphylla*	Mean	18.65	21.80	Small	0.86
	Stdev	1.18	0.90		
	Min	16.60	20.30		
	Max	20.80	23.60		
	CV	6.32	4.13		
*C. dalmaticum*	Mean	26.67	32.47	Medium	0.82
	Stdev	1.52	3.28		
	Min	24.40	27.50		
	Max	29.20	37.90		
	CV	5.70	10.10		
*C. dalmaticum* (syn. *M. bulgarica*)	Mean	21.37	25.95	Medium	0.82
	Stdev	2.82	1.35		
	Min	15.70	23.40		
	Max	24.10	27.30		
	CV	13.20	5.20		
*C. frivaldszkyanum*	Mean	25.46	28.36	Medium	0.90
	Stdev	1.23	0.99		
	Min	24.00	26.30		
	Max	28.50	29.40		
	CV	4.83	3.49		
*C. pulegium*	Mean	26.87	28.90	Medium	0.93
	Stdev	2.54	3.00		
	Min	25.00	26.10		
	Max	21.00	33.60		
	CV	9.45	10.38		
*C. serpyllifolium*	Mean	22.27	26.21	Medium	0.85
	Stdev	3.10	2.07		
	Min	12.60	22.60		
	Max	26.30	29.60		
	CV	13.92	7.89		
*C. thymifolium*	Mean	22.15	23.52	Small	0.94
	Stdev	1.21	1.17		
	Min	19.70	21.50		
	Max	24.60	25.90		
	CV	5.46	4.97		

**Table 4 plants-10-01666-t004:** Results of Tukey HSD post hoc test at the 0.05 level for the polar length of studied *Micromeria* and *Clinopodium* taxa. Presence of asterisk (*) indicates significance at *p* ≤ 0.05.

Taxon	Mc4	McK	Mcr2	McrP1	Mg1	MgF
McK	0.996467					
Mcr2	0.000029 *	0.000029 *				
McrP1	0.000029 *	0.000029 *	0.999995			
Mg1	0.000029 *	0.000029 *	0.000041 *	0.000605 *		
MgF	0.000029 *	0.000029 *	0.981839	0.999995	0.013724 *	
Mj1	0.054533	0.000335 *	0.000059 *	0.000030 *	0.000029 *	0.000029 *
Mk1	0. 000813 *	0.000030 *	0.005354 *	0.000199 *	0.000029 *	0.000032 *
Ml1	0.000029 *	0.000032 *	0.000029 *	0.000029 *	0.000029 *	0.000029 *
Mm	0.000030 *	0.000389 *	0.000029 *	0.000029 *	0.000029 *	0.000029 *
Cd	0.000029 *	0.000029 *	0.992281	1.000000	0.008540 *	1.000000
CdB1	1.000000	0.999997	0.000029 *	0.000029 *	0.000029 *	0.000029 *
Cf1	0.000029 *	0.000029 *	0.999806 *	0.930688	0.000029 *	0.464403
Cp	0.000029 *	0.000029 *	0.919882 *	0.999711	0.037175 *	1.000000
Cs	0.998023	0.469331	0.000029	0.000029 *	0.000029 *	0.000029 *
Ct4	0.999848	0.649443	0.000029	0.000029 *	0.000029 *	0.000029 *
**Taxon**	**Mj1**	**Mk**	**Ml1**	**Mm**		**Cd**
Mk1	0.999474					
Ml1	0.000029 *	0.000029 *				
Mm	0.000029 *	0.000029 *	0.999931			
Cd	0.000029 *	0.000035 *	0.000029 *	0.000029 *		
CdB1	0.007514 *	0.000077 *	0.000029 *	0.000037 *		0.000029 *
Cf1	0.003785 *	0.151169	0.000029 *	0.000029 *		0.559547
Cp	0.000029 *	0.000030 *	0.000029 *	0.000029 *		1.000000
Cs	0.711451	0.081441	0.000029 *	0.000029 *		0.000029 *
Ct4	0.534318	0.038831 *	0.000029 *	0.000029 *		0.000029 *
**Taxon**	**Cd**	**CdB1**	**Cf1**	**Cp**		**Cs**
CdB1	0.000029 *					
Cf1	0.559547	0.000029 *				
Cp	1.000000	0.000029 *	0.272970			
Cs	0.000029 *	0.924334	0.000029 *	0.000029 *		
Ct4	0.000029 *	0.977758	0.000029 *	0.000029 *		1.000000

**Table 5 plants-10-01666-t005:** Results of Tukey HSD post hoc test at the 0.05 level for the equatorial diameter of pollen grain of studied *Micromeria* and *Clinopodium* taxa. Presence of asterisk (*) indicates significance at *p* ≤ 0.05.

Taxon	Mc4	McK	Mcr2	McrP1	Mg1	MgF
McK	0.000032 *					
Mcr2	0.001274 *	0.000029				
McrP1	0.000029 *	0.000029 *	0.324744			
Mg1	0.000029 *	0.000029 *	0.000029 *	0.000029 *		
MgF	0.000029 *	0.000029 *	0.000029 *	0.000029 *	0.983544	
Mj1	0.638719	0.000029 *	0.763081	0.000190 *	0.000029 *	0.000029 *
Mk1	0.012601 *	0.000029 *	1.000000	0.080933	0.000029 *	0.000029 *
Ml1	0.000029 *	0.044219 *	0.000029 *	0.000029 *	0.000029 *	0.000029 *
Mm	0.000029 *	0.725134	0.000029 *	0.000029 *	0.000029 *	0.000029 *
Cd	0.000029 *	0.000029 *	0.000029 *	0.000039 *	0.003949	0.379663
CdB1	1.000000	0.000031 *	0.001974 *	0.000029	0.000029 *	0.000029 *
Cf1	0.000381 *	0.000029 *	1.000000	0.518758	0.000029 *	0.000029 *
Cp	0.000031 *	0.000029 *	0.996257	0.990120	0.000029 *	0.000029 *
Cs	1.000000	0.000029 *	0.014167 *	0.000029 *	0.000029 *	0.000029 *
Ct4	0.000793 *	0.999419	0.000029 *	0.000029 *	0.000029 *	0.000029 *
**Taxon**	**Mj1**	**Mk**	**Ml1**	**Mm**	**Cd**	
Mk1	0.977496					
Ml1	0.000029 *	0.000029 *				
Mm	0.000029 *	0.000029 *	0.995845			
Cd	0.000029 *	0.000029 *	0.000029*	0.000029 *		
CdB1	0.711996	0.018248 *	0.000029 *	0.000029 *	0.000029 *	
Cf1	0.566977	0.999982	0.000029 *	0.000029 *	0.000029 *	
Cp	0.055158	0.897260	0.000029 *	0.000029 *	0.000029 *	
Cs	0.950327	0.090359	0.000029 *	0.000029 *	0.000029 *	
Ct4	0.000029 *	0.000029 *	0.000581 *	0.083981	0.000029 *	
**Taxon**	**Cd**	**CdB1**	**Cf1**	**Cp**	**Cs**	
CdB1	0.000029 *					
Cf1	0.000029 *	0.000598 *				
Cp	0.000029 *	0.000032 *	0.999755			
Cs	0.000029 *	1.000000	0.004970 *	0.000068 *		
Ct4	0.000029 *	0.000505 *	0.000029 *	0.000029 *	0.000068 *	

**Table 6 plants-10-01666-t006:** Details on origin and collection data of investigated *Micromeria* (*M*.) and *Clinopodium* (*C*.) taxa.

Taxa—According to		Collecting Site	Voucher no.
Bräuchler et al., 2008 (Code)	Balkan Literature		
*M. cristata* ssp. *cristata* (Mc1)	*M. cristata*	Humsko brdo Mt (Serbia)	HFK-HR-51126
*M. cristata* ssp. *cristata* (Mc2)	*M. cristata*	Vitosha Mt (Bulgaria)	HFK-HR-51132
*M. cristata* ssp. *cristata* (Mc3)	*M. cristata*	Pirin Mt (Bulgaria)	HFK-HR-51133
*M. cristata* ssp. *cristata* (Mc4)	*M. cristata*	Demir Kapija (N. Macedonia) *	HFK-HR-51141
*M. cristata* ssp. *cristata* (Mc5)	*M. cristata*	Nomos Serron (Greece)	HFK-HR-51145
*M. cristata* ssp. *cristata* (Mc6)	*M. cristata*	Paikon Mt (Greece)	HFK-HR-51146
*M. cristata* ssp. *kosaninii* (McK)	*M. kosaninii*	Pletvar (N. Macedonia) *	HFK-HR-51142
*M. croatica* (Mcr1)	*M. croatica*	Rossijev kuk (Croatia)	HFK-HR-51016
*M. croatica* (Mcr2)	*M. croatica*	Stupačinovo (Croatia) *	HFK-HR-51017
*M. croatica* (Mcr3)	*M. croatica*	Dubočani (BIH)	HFK-HR-51020
*M. croatica* (McrP1)	*M*. *pseudocroatica*	Pijavičino (Croatia) *	HFK-HR-51032
*M. croatica* (McrP2)	*M*. *pseudocroatica*	Prapratno (Croatia)	HFK-HR-51033
*M. graeca* ssp. *graeca* (Mg1)	*M. graeca*	Malo zlo polje (Croatia) *	HFK-HR-51036
*M. graeca* ssp. *graeca* (Mg2)	*M. graeca*	Komiža (Croatia)	HFK-HR-51037
*M. graeca* ssp. *fruticulosa* (MgF)	*M. fruticulosa*	Sušac Island (Croatia) *	HFK-HR-51038
*M. juliana* (Mj1)	*M. juliana*	Grude (BIH) *	HFK-HR-51043
*M. juliana* (Mj2)	*M. juliana*	Cijevna Canyon (Montenegro)	HFK-HR-51045
*M. juliana* (Mj3)	*M. juliana*	Rajec Reka (N. Macedonia)	HFK-HR-51168
*M. juliana* (Mj4)	*M. juliana*	Cholomon Mt (Greece)	HFK-HR-51147
*M. kerneri* (Mk1)	*M. kerneri*	Zavratnica (Croatia) *	HFK-HR-51018
*M. kerneri* (Mk2)	*M. kerneri*	Mostar (BIH)	HFK-HR-51044
*M. longipedunculata* (Ml1)	*M. parviflora*	Jazina (BIH) *	HFK-HR-51046
*M. longipedunculata* (Ml2)	*M. parviflora*	Krivošije Mt (Montenegro)	HFK-HR-51047
*M. microphylla* (Mm)	*M. microphylla*	Svetac Island (Croatia) *	HFK-HR-51039
*C. dalmaticum* (Cd)	*M. dalmatica*	Orjen Mt (Montenegro) *	HFK-HR-51051
*C. dalmaticum* (CdB1)	*M. bulgarica*	Mesta River Valley (Bulgaria) *	HFK-HR-51134
*C. dalmaticum* (CdB2)	*M. bulgarica*	Vlahina Mt (Bulgaria)	HFK-HR-51135
*C. frivaldszkyanum* (Cf1)	*M. frivaldszkyana*	Malusha peak (Bulgaria) *	HFK-HR-51136
*C. frivaldszkyanum* (Cf2)	*M. frivaldszkyana*	Vikanata Skala Nature Monument (Bulgaria)	HFK-HR-51137
*C**. pulegium* (Cp)	*M* *. pulegium*	Međeđa (BIH) *	HFK-HR-51050
*C. serpyllifolium* (Cs)	*M. albanica*	Prizren (Kosovo) *	HFK-HR-51074
*C. thymifolium* (Ct1)	*M. thymifolia*	Učka Mt (Croatia)	HFK-HR-51077
*C. thymifolium* (Ct2)	*M. thymifolia*	Dokozina plan (Croatia)	HFK-HR-51082
*C. thymifolium* (Ct3)	*M. thymifolia*	Šušanj (Croatia)	HFK-HR-51081
*C. thymifolium* (Ct4)	*M. thymifolia*	Blidinje (BIH) *	HFK-HR-51042
*C. thymifolium* (Ct5)	*M. thymifolia*	Manastir Morača (Montenegro)	HFK-HR-51053

Note: BIH = Bosnia and Herzegovina; N. = North; * = site of pollen collecting.

**Table 7 plants-10-01666-t007:** Details on collection data of investigated *Micromeria* (*M*.) and *Clinopodium* (*C*.) taxa.

Taxa–According to		Latitude	Longitude	Altitude a.s.l. (m)
Bräuchler et al., 2008 (Code)	Balkan Literature			
*M. cristata* (Mc1)	*M. cristata*	43°22′45.0″	21°53′50.1″	387
*M. cristata* (Mc2)	*M. cristata*	42°29′33.4″	23°11′43.1″	980
*M. cristata* (Mc3)	*M. cristata*	41°46′11.4″	23°25′19.1″	1970
*M. cristata* (Mc4)	*M. cristata*	41°24′18.1″	22°15′47.0″	111
*M. cristata* (Mc5)	*M. cristata*	41°15′10.3″	23°24′49.8″	190
*M. cristata* (Mc6)	*M. cristata*	40°57′21.2″	22°20′02.0″	1650
*M. cristata* ssp. *kosaninii* (McK)	*M. kosaninii*	41°22′09.0″	21°39′06.1″	1020
*M. croatica* (Mcr1)	*M. croatica*	44°45′51.1″	4°59′17.1″	1641
*M. croatica* (Mcr2)	*M. croatica*	44°32′37.5″	15°10′04.7″	1058
*M. croatica* (Mcr3)	*M. croatica*	43°35′10.1″	18°04′44.0″	715
*M. croatica* (McrP1)	*M*. *pseudocroatica*	42°56′58.8″	17°22′02.1″	436
*M. croatica* (McrP2)	*M*. *pseudocroatica*	42°49′28.1″	17°39′57.9′	167
*M. graeca* (Mg1)	*M. graeca*	43°03′43.3′‘	16°12′55.9″	137
*M. graeca* (Mg2)	*M. graeca*	43°02′17.4″	16°05′49.5″	43
*M. graeca* ssp. *fruticulosa* (MgF)	*M. fruticulosa*	43°02′13.4″	16°05′50.5″	10
*M. juliana* (Mj1)	*M. juliana*	43°23′18.5″	17°23′27.8″	495
*M. juliana* (Mj2)	*M. juliana*	42°25′44.6″	19°28′53.5″	157
*M. juliana* (Mj3)	*M. juliana*	41°26′12.2″	21°52′06.4″	199
*M. juliana* (Mj4)	*M. juliana*	40°27′30.0″	23°31′04.6″	1100
*M. kerneri* (Mk1)	*M. kerneri*	44°42′02.2″	14°54′45.6″	161
*M. kerneri* (Mk2)	*M. kerneri*	43°20′27.7″	17°48′53.2″	61
*M. longipedunculata* (Ml1)	*M. parviflora*	42°42′11.7″	18°30′37.1″	347
*M. longipedunculata* (Ml2)	*M. parviflora*	42°32′51.8″	18°42′41.1″	624
*M. microphylla* (Mm)	*M. microphylla*	43°01′07.8″	15°45′08.2″	28
*C. dalmaticum* (Cd)	*M. dalmatica*	42°33′45.1″	18°37′36.6″	1074
*C. dalmaticum* (CdB1)	*M. bulgarica*	41°40′46.6″	23°43′29.7″	580
*C. dalmaticum* (CdB2)	*M. bulgarica*	41°50′30.6″	22°59′27.8″	1140
*C. frivaldszkyanum* (Cf1)	*M. frivaldszkyana*	42°45′01.0″	25°16′53.7′	1311
*C. frivaldszkyanum* (Cf2)	*M. frivaldszkyana*	42°45′57.6″	25°30′08.1″	1040
*C**. pulegium* (Cp)	*M* *. pulegium*	43°44′01.2″	19°17′07.3″	563
*C. serpyllifolium* (Cs)	*M. albanica*	42°12′01.6″	20°45′57.7″	376
*C. thymifolium* (Ct1)	*M. thymifolia*	45°17′08.1″	14°12′02.4″	1189
*C. thymifolium* (Ct2)	*M. thymifolia*	44°39′04.3″	15°02′39.1″	1441
*C. thymifolium* (Ct3)	*M. thymifolia*	44°31′33.8″	15°06′45.1″	604
*C. thymifolium* (Ct4)	*M. thymifolia*	43°31′07.8″	17°23′09.8″	1195
*C. thymifolium* (Ct5)	*M. thymifolia*	42°45′50.9″	19°23′34.6″	301

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
