# Peer review of "Micromorphological Traits of Balcanic Micromeria and Closely Related Clinopodium Species (Lamiaceae)"

_plants, 2021, doi:10.3390/plants10081666_

Round 1

Reviewer 1 Report

The manuscript deals with trichomes and pollen charecteristics of Micromeria and Clinopodium taxa from the Balkans. The topic is interesting. The amount of collected data important. The English is ok. The results are clearly presented and well discussed. I suggest acceptance after the following minor revisions:

Title: I suggest to delete “Benth.” and “L.” from the title

Line 27: “trichomes types and distribution” instead of “types and distribution of trichomes”

Line 28: Add “ssp. cristata”: M. cristata ssp. cristata, M. cristata ssp. kosaninii

Line 28: ssp. NOT in italics

Lines 28-29: Add “ssp. graeca”: M. graeca ssp. graeca, M. graeca ssp. fruticulosa

Line 39: Pseudomelissa in italics

Keywords: Do not repeat in the keywords words already included in the title, i.e. delete “Clinopodium” and “Micromeria” from the Keywords

Line 44: I suggest “eudicots” instead of “dicotyledons”

Line 45: Add a reference after “more than 240 genera and more than 7200 species”

Line 52: “form” instead of “forms”

Line 58: “transferred the taxa included in section Pseudomelissa” instead of “transferred the section Pseudomelissa”

Line 77: Did you include all Clinopodium species occurring in the Balkans? Or only those transferred from Micromeria sect. Pseudomelissa to Clinopodium?

Line 81: “from Micromeria sect. Pseudomelissa” instead of “from section Pseudomelissa”

Line 96: “Tables” instead of “Table”

Lines 106-109: Names of taxa all in italics. In addition, add “ssp. cristata” to “Micromeria cristata” and “ssp. graeca” to “M. graeca”

Line 112: Micromeria in italics

Line 113: Clinopodium in italics

Line 113: Clinopodium serpyllifolium in italics

Line 114: Micromeria and Clinopodium in italics

Line 116: Clinopodium serpyllifolium and sirnakense in italics

Line 117: Clinopodium in italics

Lines 117-118: Micromeria in italics

Line 119: C. frivaldszkyanum, C. pulegium and C. frivaldszkyanum in italics

Line 120: Micromeria in italics

Line 121: Clinopodium in italics

Line 122: Micromeria in italics

Lines 122-123: Clinopodium in italics

Line 127: “Tables” instead of “Table”

Line 130: M. croatica and M. longipedunculata in italics

Line 131: Micromeria fruticosa in italics

Lines 131-132: “Boiss. et Hohen.” not in italics

Line 132: M. croatica, M. kerneri, M. juliana and M. longipedunculata in italics

Line 132: Delete “Murb.”

Lines 133-134: Clinopodium serpyllifolium in italics

Line 134: C. dalmaticum and C. thymifolium in italics

Line 135: C. thymifolium in italics

Line 136: C. frivaldszkyanum in italics

Line 142: Micromeria in italics

Line 143: Clinopodium in italics

Line 144: Micromeria croatica in italics

Lines 146-148: Names of all taxa in italics

Line 148: Delete “Non-glandular trichomes (NG)”, given that they are not shown in the Figure

Line 151: Delete “Murb.”

Lines 151-157: Names of all taxa in italics

Line 158, Caption to Table 1: Micromeria in italics

Table 1: add “ssp. cristata” (6-times) to “Micromeria cristata” and “ssp. graeca” (2-times) to “M. graeca”

Lines 212-216: Names of all taxa in italics

Line 223: Micromeria in italics

Line 225, Caption to Table 2: Clinopodium in italics

Line 248: The first part is missing. It begins with “chomes; trichomes frequency”

Lines 250-253: Names of all taxa in italics (apart from Lamiaceae)

Line 251: Delete “Murb.”

Lines 257-261: Names of all taxa in italics

Line 266: Clinopodium in italics

Lines 268-277: Names of all taxa in italics

Line 276 Add the author(s) to Majorana syriaca

Line 278: Micromeria in italics

Line 279: Clinopodium in italics

Line 282: “Figures” instead of “Figure”

Line 286: Add “ssp. cristata”: M. cristata ssp. cristata

Line 287: Add “ssp. graeca”: M. graeca ssp. graeca

Line 300: Add “ssp. cristata”: M. cristata ssp. cristata

Line 312: Add “ssp. graeca”: M. graeca ssp. graeca

Table 3: Add “ssp. cristata” to “M. cristata” and “ssp. graeca” to “M. graeca”

Line 334: “Tables” instead of “Table”

Line 335: Micromeria and Clinopodium in italics

Line 336: Add “ssp. cristata” to “M. cristata”

Line 338: Add “ssp. graeca”: M. graeca ssp. graeca

Lines 336-341: Names of all taxa in italics

Line 360: “M. cristata (Hampe) Griseb. ssp. cristata” instead of “M. cristata (Hampe) Griseb.”

Line 362: “M. graeca (L.) Benth. ex Rchb. ssp. graeca” instead of “M. graeca (L.) Benth. ex Rchb.”

Line 370: Delete the comma after “M. pulegium (Rochel) Benth.,”

Table 6, first column: add “ssp. cristata” (2x6-times) to “M. cristata” and “ssp. graeca” (2x2-times) to “M. graeca”

Line 403: Add a reference

Figure 5, Add to the caption: Micromeria cristata ssp. cristata (Mc), M. cristata ssp. kosaninii (McK), M. croatica (Mcr), etc etc.

Line 411: “Micromeria and Clinopodium taxa” instead of “Micromeria and Clinopodium species”

Line 418: “characteristic of species” instead of “characteristic pf species”

Author Response

Dear Reviewer,

Reviewer 2 Report

Dear all, 

Please see my comments and suggestion in the file attached. I also recommend to separate Results and Discussion, and the main conclusions should be expressed more clearly.

Best wishes,

Alexander P. Sukhorukov

Author Response

Dear Reviewer,

Sincerely, 

Valerija Dunkić 

Reviewer 3 Report

The paper submitted by Kramer et al is focused on the evaluation of morphological differences between some Micromeria and Clinopodium species. The manuscript is well written and presents some interesting observations. The manuscript requires some minor checks and should be published in Plants.

1) please increase the quality of fig 5 - in its present form it is not readable for the reader and could be deleted from the main text

2) it would be useful for the reader to present photos of investigated plant materials as whole plants

3) is it possible to differentiate between investigated species based on observed features? maybe it would be possible to create a decision tree for the identification of investigated species?

4) table 4 - it is not clear what pluses and minuses mean - what is considered as rare or abundant? it should be clearly explained

5) Latin names of all species should be given in italics in the whole manuscript

6 ) is it possible to use quantitative data to draw PCA plots for analysed samples?

Author Response

(The authors gave the same response as above.)
